# Fecal glucocorticoid metabolite levels in captive Indian leopards *(Panthera pardus fusca)* housed under three different enrichment regimes

**Nirali Panchal, Chena Desai, Ratna Ghosal** ✪ *

Biological and Life Sciences, School of Arts and Sciences, Ahmedabad University, Ahmedabad, Gujarat, India

* ratna.ghosal@ahduni.edu.in

**Data Availability Statement:** All relevant data are within the paper and its Supporting Information files.

## Abstract

Environmental enrichment improves the health and wellbeing of zoo animals. To test this hypothesis, we used Indian leopards (*Panthera pardus fusca*), one of the popular zoo animals, as a model organism to understand effects of active and passive enrichment elements on stress hormone levels of captive individuals. We included three enrichment categories, category 'A' (having both active: cage size of 1204 $m^3$ with raised platforms and earthen flooring, and passive: controlled temperature, playback of forest sounds and sound proof glass to filter visitors' noise, enrichment elements), category 'B' (active enrichment type I, cage size of 264 $m^3$ with air coolers), and category C (active enrichment type II, cage size of 517 $m^3$ without air coolers) for leopards (n = 14) housed in two Indian zoos. We used a group-specific enzyme immunoassay to measure fecal glucocorticoid metabolites (fGCM) in captive leopards. For comparison, we analysed samples from free-ranging leopards, as well. fGCM levels (Mean±SEM) were 10.45±2.01 and 0.95±0.003 µg/g dry feces in captive and free-ranging leopards, respectively. Our results revealed that fGCM levels of leopards in categories B and C were significantly ($P<0.05$) different from each other, thus, indicating cage size (an active enrichment element) as an important factor in influencing the physiology of the sampled animals. Overall, the findings of the study will contribute towards informed policies for management of captive Indian leopards.

## Introduction

Conservation action requires sustainable management of captive and zoo-housed individuals of a species. Globally, there are approximately 10,000 zoos and captive breeding centers, and India alone has more than 145 zoological gardens [1]. Most of the natural populations of animals are declining due to deforestation and urbanization; thus, zoo or captive populations provide an alternative source of genetic materials for vulnerable or threatened species [2]. Besides conservation research, zoos are also a source of recreation, entertainment and education for the general public [3, 4]. Thus, as a part of the *ex-situ* conservation action plan, monitoring the health and wellbeing

**Funding:** Dr. Ratna Ghosal received grant AU/SUG/ SAS/BLS/2018-19/20 from Ahmedabad University. The funders had no role in study design, data collection and analysis, decision to publish, or preparation of the manuscript.

**Competing interests:** The authors have declared that no competing interests exist.

of zoo animals becomes a priority for ecologists and conservation managers, as well. However, being away from their natural habitat, zoo animals often suffer (mentally and physically) within the artificial, unfamiliar, captive environment [5, 6]. Thus, to maintain the wellbeing of the zoo animals, several types of physical and virtual stimuli need to be added to the captive environment with an attempt to supplement the captivity with enriched habitat conditions.

Enrichment is the alteration made to the environment of captive animals for their physiological and psychological wellbeing. Studies have shown that enrichment can be divided into two categories, active and passive. Active enrichment can be defined as enrichment that requires animals to perform some sort of physical activity or the organisms are in direct contact with a physical object [7]. For example, Carlstead and Shepherdson [8] showed that providing structural enrichment that mimicked housing conditions similar to the natural setup, increased success rate of reproduction in animals like rats (*Rattus rattus*), ferrets (*Mustela putorius*), great apes and ungulates. Similarly, stereotypic behaviour was reduced in captive bears (*Ursus americanus*) when food presentation was varied [5]. For example, when food was provided in a log filled with honey and was hidden throughout the exhibit, bears reduced stereotypic pacing from 125min/day to 20min/day, and increased the rate of their exploratory behaviours [5]. On the other hand, passive enrichment can be defined as a regime that does not include any interaction with a physical structure or any kind of direct contact with a living being [7]. Passive enrichment includes visual, olfactory, and auditory enrichment, and mostly consists of modifications that enrich the ambient environment without necessarily involving any physical interaction. Television, video and computer games, mirror and colors are known to be effective as visual, passive enrichment for rhesus macaques (*Macaca mulatta*), horses (*Equus ferus*) and chimpanzees (*Pan troglodytes*) [9]. In the case of auditory enrichment, sounds specific to animal's natural environment, and even classical music, radio broadcasts and instrumental music acted as enrichment to female African leopards (*Panthera pardus pardus*), Asian elephants (*Elephas maximus*), gorillas (*Gorilla gorilla*), Guinea pigs (*Cavia porcellus*) and rats [9]. Management of zoo animals is challenging due to vast differences between the natural and the captive environment, and thus, captive conditions often trigger physiological stress reactions, which may eventually impact an animal's wellbeing [10, 11].

Stress can be defined as loss of homeostasis, and a stressor is an event or force, which causes this disruption [12]. A series of physiological events take place inside the body to restore homeostasis, a way in which the body responds towards the stressor. Stress can be caused by physical factors, for example injuries, conflict [13], environmental factors such as temperature, humidity, sunlight, as well as internal factors such as anoxia, hypoglycemia [14]. Stressful conditions may also impact animal behavior, for example, causing increased aggression or sign of submissive behavior under such conditions [14]. Most kinds of stressors trigger the release of glucocorticoids (GCs) from adrenal glands [14]. GCs are known to increase blood glucose levels, suppress the immune system to help maintain homeostasis, and support metabolism of fats, proteins and carbohydrates [15, 16] to mobilize energy under stressful situations. The GC spike in the bloodstream can be measured by routine hormone assays, and several of GC's metabolites can be measured in other biological samples, for example, feces, urine, mucus [17]. Studies have shown that captive conditions are highly stressful for most animals and often such artificial environments may lead towards poor reproductive performance and/or higher prevalence of diseases among the zoo populations [18, 19].

Confinement, artificial environment, visitors [20, 21] and isolation of social animals [22] are the key factors known to cause high levels of stress in captive animals. Thus, to improvise *ex-situ* conservation efforts, enriched habitats are provided to captive animals and their physiological response towards the enrichment can be assessed through measurements of stress hormones, mostly by monitoring levels of GCs [9]. For example, a study on small felids

(*Leopardus tigrinus* and *Leopardus wiedii*) showed that females when provided with bigger enclosures, an example of active enrichment, had reduced stress hormone levels, and resumed reproductive cyclicity when compared to females maintained under non-enriched, stressful conditions (small and barren enclosures) [23]. Another study on Clouded leopards (*Neofelis nebulosa*) showed that certain unusual behaviours, for example, fur plucking and nail biting, were relatively higher in captive animals, thus indicating stressful conditions [24]. To reduce such behaviours, different types of enrichment (active and passive) were provided to captive Clouded leopards [24], of which increased time spent with the keeper (a type of active enrichment element) was found to be effective in reducing high fecal glucocorticoid metabolite (fGCM) levels in Clouded leopards. Globally, a large population of different species is present in captivity, and with an aim to provide improvised husbandry practices; studies need to be conducted to understand the effect of diverse enrichment elements (active or passive or a combination of both) on the physiological wellbeing of captive populations.

India is home to a rich biodiversity, but hosts a large population of captive animals, as well. Monitoring wellbeing of the captive animals is one of the priorities for the managers and the veterinarians of the Indian zoos. One such popular animal housed in a large number of Indian zoos is the Indian Leopard (*Panthera pardus fusca*), categorized as endangered as per the IUCN Red List (2016). Out of 145 zoos in India, leopards are present in 76 different zoos across the country [25], and each zoo having approximately 8–10 leopards. Studies have shown that captivity is very stressful for leopards and induces all sorts of unusual behaviours. For example, Indian leopards kept in zoos in 3 different states of India (Maharashtra, Kerala and Delhi) showed high levels of stereotypic behaviours (repetitive walks, chewing paws and snapping) and had higher fGCM levels under conditions with no enrichment [26]. However, fGCM levels and intensity of stereotypic behaviours decreased when animals were provided with different types of structural enrichment, for example, den or pool within an enclosure, and more time spent with a keeper providing care. Another study showed that captive Indian leopards preferred to spend more time in enriched zones having structural elements, for example, trees, barrels, sleeping platform and logs, and had decreased rates of stereotypic pacing under such conditions [27]. However, both studies on the captive Indian leopards included only components of active enrichment. To the best of our knowledge, so far no study has been conducted to assess the effects of passive enrichment components (auditory, visual or olfactory stimuli) on the physiological wellbeing of captive Indian leopards, and thus, warrants future investigation.

In this study, we monitored the impact of three different enrichment conditions (referred to as categories 'A', 'B' and 'C') on the stress physiology of the zoo leopards. The first category (category 'A') had both active and passive elements, whereas active enrichment type-I (category 'B') and -II (category 'C') had active enrichment elements only. This is the first study to include both active and passive enrichment elements, and assess their effects on the physiological stress levels of the Indian leopards. To measure physiological responses towards a particular enrichment regime, we utilized an enzyme immunoassay to measure fGCM levels in captive Indian leopards. For comparison using the same assay, fGCM concentrations were also measured in scat samples collected from free-ranging leopards. The overarching goal of our study is to develop an improved protocol for better management practices for the target species.

## Materials and methods

### Study site and animals for captive sampling

Sampling was carried out in 14 leopards (8 males and 6 females) (Table 1). Four male leopards and four females were from Kankaria zoo, Ahmedabad, located in the southeastern part of the city, while remaining four males and two females were from Sayajibaug zoo, Baroda (all in the

**Table 1. Details on enrichment regimes for leopards in Kankaria and Baroda zoos, Gujarat.**

| Zoo name | Enrichment category | Number of animals; Sex | Enrichment details | Type of food provided, feeding frequency | Feeding time |
|---|---|---|---|---|---|
| Kankaria | Category 'A', indoor facility with both active and passive enrichment elements | N = 4; Females | • Artificial light condition<br>• Playback of natural, forest sounds<br>• Reverse day and night cycle<br>• Sound proof glasses to filter visitors' noise<br>• Raised platforms with stairs<br>• Earthen flooring with controlled temperature at 25°C<br>• Cage size of 1204 m$^3$ | 3.6–4 kg buffalo meat/individual. Daily | 9:00 A.M. |
| | Category 'B', outdoor facility with active enrichment type I | N = 4; Males | • Cages with cemented and earthen flooring<br>• Raised platforms with stairs<br>• Air coolers during summer<br>• Cage size of 264 m$^3$ | 3.6–4 kg buffalo meat/individual. Daily except on Fridays | 9:30–10:00 A.M. |
| Baroda | Category 'C', outdoor facility with active enrichment type II | N = 6; 4 males and 2 females | • Cages with cemented and earthen flooring<br>• Raised platforms with stairs<br>• Cages maintained at ambient temperature across all seasons<br>• Cage size of 517 m$^3$ | 3.6–4 kg buffalo meat / individual. Daily | 9:30–10:00 A.M. |

state of Gujarat, India; Fig 1). Kankaria/Kamala Nehru zoo in Ahmedabad city, Gujarat had two different housing conditions for mammals (indoor and outdoor, Table 1). Sayajibaug zoo (referred to as Baroda zoo from here onwards) in Baroda city, Gujarat, had only one type of housing condition, the outdoor (Table 1). Both zoos are only 72 miles apart from each other, and experience a similar weather pattern in terms of temperature and rainfall. Leopards in both the zoos were adults (Table 1) and were maintained under a similar diet regime. From now onwards, we will refer to the housing conditions in two zoos as category 'A' for Kankaria indoor, category 'B' for Kankaria outdoor, and category 'C' for Baroda outdoor. Table 1 gives details on active and passive enrichment elements that were provided to the leopards within each of these categories, and the number of leopards maintained under these conditions. In all three categories, leopards were housed in individual cages. Since leopards use the height of a cage for climbing and jumping activities, we calculated cage size as a product of length, breadth and height of the cage. Category 'A' had the highest enrichment, having the largest (1204 m$^3$) cage size when compared to the other two categories, and had several active (earthen flooring and raised platforms) and passive (sound proof glass to filter visitors' noise, controlled temperature and playback of natural, forest sounds) elements for enrichment (Table 1). Category 'A' was the only regime that had passive enrichment elements for the leopards. Out of all the three categories, category 'B' had the smallest cage size (264 m$^3$) and was provided with a few active, structural enrichment elements, for example, air coolers during summer, earthen flooring and raised platforms for climbing. Category 'C' had medium size cages (517 m$^3$) but had no coolers, and was provided with similar enrichment elements as category B, having earthen floors and raised platforms (Table 1). The sampled leopards were either captive-born or captured from the wild, and time since captivity varied from 3 to 18 years (Table 2). The detailed information on each of the sampled leopards under each housing category is provided in Table 2.

For all the three categories, summer sampling (with ambient temperature ranging from 39–42°C) [28] was conducted from June to July 2019 and winter sampling (with ambient

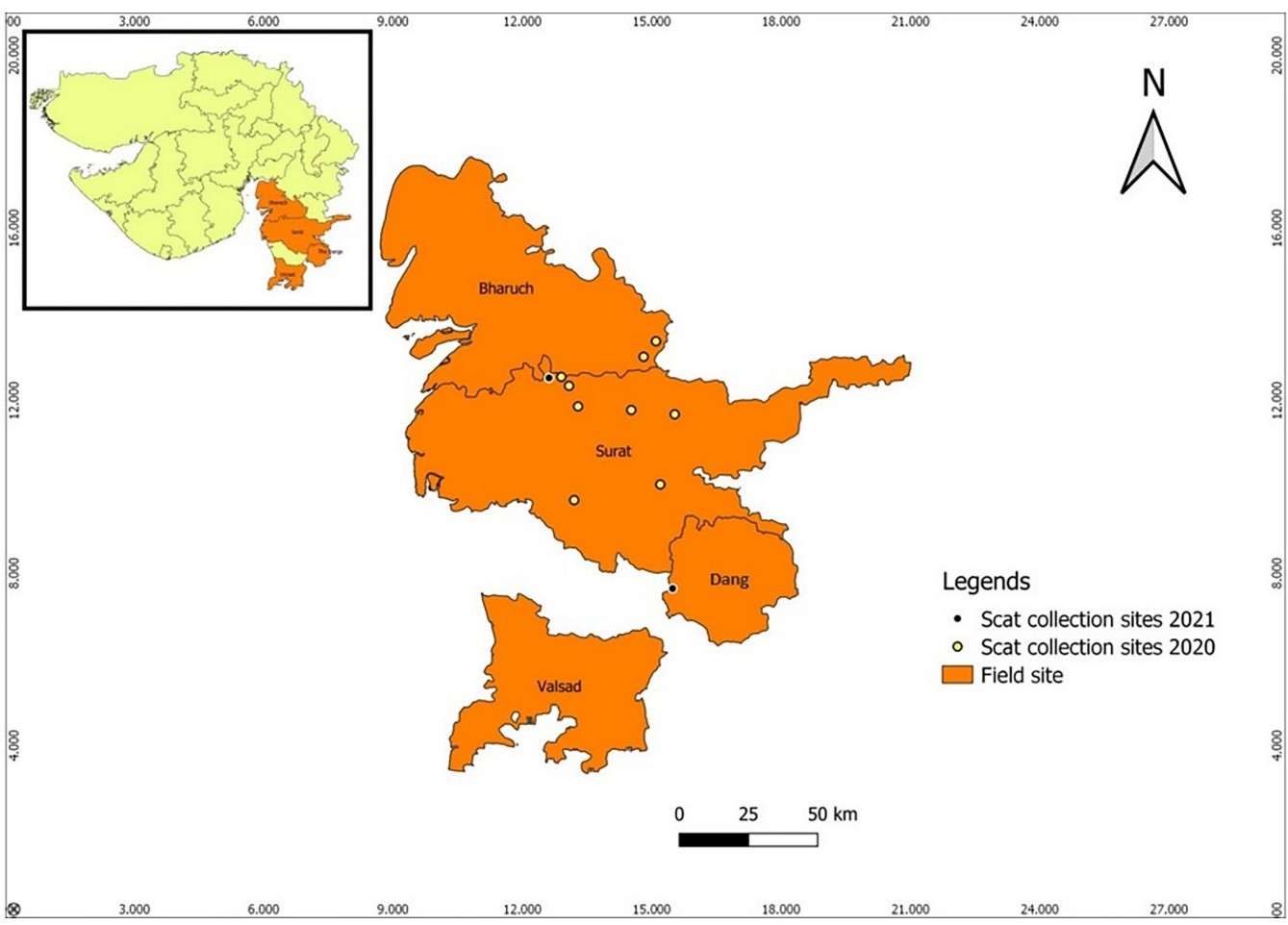

**Fig 1. Map showing the study sites for captive and free-ranging populations of the Indian leopards.** Inset panel shows the entire map for the state of Gujarat, India, with approximate locations of Kankaria and Baroda zoos, the study sites for captive leopard populations. Main panel shows the scat collection sites for free-ranging leopards in South Gujarat.

temperature ranging from 28–31˚C) [29] was carried out from December to February 2020. In total, 119 samples were collected from 14 leopards (n = 68 samples for males and n = 52 samples for females) across the three categories. On an average, we collected 3–7 scats per individual per season. Category 'A' had a total of 12 and 21 samples collected during summer and winter seasons, respectively. Accordingly, category 'B' had a total of 14 and 13 samples, and category 'C' had 36 and 31 total scats, collected during summer and winter seasons, respectively. Due to defecation in water or defecating at an inaccessible spot within the cage, we were not able to collect samples from a few individuals during certain weeks. Samples (~5 g) were collected using ice-cream sticks and zip lock bags. After collection, samples were stored at -20˚C until analysis. All necessary permits and ethics approval were obtained from the concerned authorities.

## Study site for sampling free-ranging leopards

Four districts, namely, Surat, Bharuch, Valsad and Dang within the southern part of Gujarat state, India, were chosen as the field sites for sampling scats from free-ranging leopards. The annual rainfall in south Gujarat is 900–2800 mm, and the average maximum temperature in

**Table 2. Detailed information on sampled individuals at both Kankaria and Baroda zoos, Gujarat.**

| Animal Name (Zoo name, housing condition) | Place of Birth and date of arrival to respective zoos | Sex | Age at rescue (years) | Approx. age at sampling time (years) | Remarks |
|---|---|---|---|---|---|
| Mili (Category 'A') | Rescued from Dang, Valsad district, Gujarat Date: 05/03/2008 | Female | 7 | 19 | Died on 08/09/2020 due to old age |
| Jaya (Category 'A') | Rescued from Borivali, Maharashtra Date:29/09/2005 | Female | 5 | 19 | Died on 24/12/2020 due to old age |
| Vanshankri (Category 'A') | Born at Shimoga zoo, Karnataka Date:09/07/2018 | Female | NA | 5.5 | Born on 30/11/2015 |
| Tunga (Category 'A') | Born at Shimoga zoo, Karnataka Date:09/07/2018 | Female | NA | 5 | Born on 30/01/2016 |
| Sunny (Category 'B') | Rescued from Borivali, Maharashtra Date:29/09/2005 | Male | 2.5 | 18 | NA |
| Pradip (Category 'B') | Rescued from Shimoga zoo, Karnataka Date:09/07/2018 | Male | 6 | 10 | NA |
| Pravin (Category 'B') | Rescued from Shimoga zoo, Karnataka Date:09/07/2018 | Male | 5 | 9 | NA |
| Sandip (Category 'B') | Born at Shimoga zoo, Karnataka Date:09/07/18 | Male | NA | 8 | Born on 23/01/2014 |
| Vishnu (Category 'C') | Captured from wild, Dhanpur Forest, Bariya Forest Division, Gujarat. Date:23/03/2009 | Male | 3 | 15 | NA |
| Naaraj (Category 'C') | Born at Baroda zoo on 04/08/2007 | Male | NA | 15 | NA |
| Neel (Category 'C') | Captured from the wild from Tejpur Forest, Assam. Date:05/03/2003 | Male | 3 | 16 | NA |
| Nayan (Category 'C') | Captured from wild, Prankad Village, Jhagadiya Taluka, Gujarat Date:01/01/2012 | Male | NA | 8 | NA |
| Naina (Category 'C') | Captured from wild, Prankad Village, Jhagadiya Taluka, Gujarat Date:01/01/2012 | Female | NA | 8 | NA |
| Netra (Category 'C') | Captured from wild, Prankad Village, Jhagadiya Taluka, Gujarat Date:01/01/2012 | Female | NA | 8 | NA |

summer goes up to 40˚C in May, which is the hottest month whereas minimum temperature lies at 26˚C. December is the coldest month of the year, and the maximum temperature averages to 25˚C and the minimum is 16˚C [30]. Overall, south Gujarat has the highest forest cover with dense canopy; wherein Dang district has a forested area of about 1,368 sq. km, Surat has an area of 496.72 sq.km, Bharuch has an area of 1142 sq.km and Valsad having a forest cover of 274.69 sq.km. [30]. According to the census in 2016, the estimated leopard population in Gujarat was about 1395 [31]. Out of which, the population count was 43 in Surat, 18 in Valsad and 43 in Dang districts of Gujarat. However, the census data for Bharuch is not available. The total population across the four districts accounts for 7.4% of the total leopard population for the state of Gujarat [31].

Scat samples were collected from Surat, Bharuch and Dang districts of Gujarat during January-March 2020, and during the month of March 2021. Though we conducted a field survey in Valsad during the month of March 2021, fresh scats were not obtained from the district. A total of 10 field hours were spent daily in tracking and collecting samples of the leopards. Only fresh scat samples were collected in a zip lock bag. Parameters like moisture content, smell and

evidence of recent activity, for example fresh tracks and pugmarks, were used as a criterion to determine freshness of the scat [32]. Scats were kept in an icebox at the field site and were transferred to -20˚C within 4–6 hours after collection. A total of 12 samples were collected randomly from fairly distant areas (80 to 140 km) across three districts of Gujarat. This was done to avoid pseudo replication by collecting repeated samples from the same individual. However, due to random sampling, the life history or the sex of the sampled leopards could not be determined. Fig 1 shows the sampling sites for free-ranging leopards within the state of Gujarat.

## Extraction and analysis of fGCMs from scat samples of both captive and free-ranging leopards

Samples were dried using a hot air oven for 24–36 hrs, until completely dry. After drying, the samples were pulverized and sieved, and the powdered sample was stored in glass bottles at room temperature until further analysis. For sample extraction, 3 ml of 80% methanol was added to 0.1 g of fecal powder. The mixture was vortexed for 3 minutes and then centrifuged at 1500 rpm for 10 minutes [33]. Supernatants were collected (1.5–2 ml) in tubes and stored at -20˚C until steroid analysis. Fecal GCMs were quantified using a 5α-pregnane-3β,11β,21-triol-20-one enzyme immunoassay (EIA) kit purchased from Rupert Palme (Department of Biomedical Sciences, University of Veterinary Medicine, Vienna). Details of the EIA, including cross-reactions are given elsewhere [34–36]. The EIA was already validated (with an ACTH challenge test) for African leopards (*Panthera pardus pardus*,) [13]. The EIA measures fGCMs with a 5α-3β,11β-diol structure [37]. To reduce non-specific binding, the assay was performed on anti-rabbit-IgG-coated (R2004, Sigma-Aldrich) microtiter plates. The assay sensitivity was 2.4 ng per gram DW. The Intra- and inter-assay coefficients of variation were 9.71% and 14.20%, respectively for an internal control sample, whose value was close to the center of the standard curve. Fecal extracts from a sample from a captive and one from a wild animal center were serially diluted and assayed to check for parallelism with the standard curve.

## Statistical analyses

To assess the effects of enrichment and season on fGCM values of captive leopards, we used linear mixed effects (LME) model [38] with maximum likelihood method. For the fixed effects, season (with two levels, summer and winter) and enrichment (with three levels, categories 'A', 'B' and 'C') were included as independent variables, and fGCM concentration was included as a dependent variable. Since the housing (category 'A') with both active and passive enrichment had only females and active enrichment type I (category 'B') had only males, sex was not included in the mixed effect analyses. Log-transformed fGCM values were included in the LME model to meet normality assumption. Each individual leopard was repeatedly sampled over time for both seasons (summer and winter), and thus, individual identity was included as a random effect within the mixed effects model. Model comparisons were conducted to arrive at the best-fit model for the given data set and post-hoc comparisons were done to explore significant interactions within the model. Out of 14 leopards in both the zoos, only four were born in captivity under zoo conditions and the other 10 were captured from the wild (Table 2). Due to the disproportionate sample size between captive and wild born leopards, we did not include the history of individuals as an independent variable in the model. Time since captivity was also not included in the model as all the individuals spent more than 5 years in captivity. Further, all the sampled individuals were adults (Table 2), thus, age was also not included as an independent variable for the mixed effects model. Adult age criterion for the leopards was followed as outlined by [39].

All analyses were conducted on R version 3.2.3 using 'nlme', 'multicomp' and 'ggplot 2' packages [38]. Post hoc multiple comparisons using Tukey's HSD were done using the R package 'emmeans'. Parallel displacement between the standard curve and serial dilutions of the fecal extract was used to biochemically validate the fGCM assay for the Indian leopards. The values within the linear range of the curve were subjected to linear regression analysis (PRISM software, version 9) using log molar concentration vs. percent antibody binding of the standard and the sample dilution curves separately. The slopes of the regression lines were compared using Student's t-test.

All the data are reported as Mean±SEM, and expressed as µg/g dry feces. Significance level was kept at P<0.05 for all the analyses.

## Results

### Fecal GCM profiles of captive leopards

Overall, fGCM levels of captive leopards were 10.45±2.01 µg/g dry feces, with male and female leopards having an average value of 10.68±2.96 and 10.13±2.71, respectively. During summer, the overall fGCM level was 5.90 ± 2.11 with leopards in 'A', 'B' and 'C' categories had values of 2.42 ± 0.6, 22.62 ± 9.12 and 1.36 ± 0.11 µg/g, respectively (Fig 2). During the winter season, the fGCM level, pooled across all the categories, was 14.92±3.32, with individuals maintained in category 'A' having a value of 25.11±6.57, those in category 'B' showed a value of 27.83±9.22 and in category 'C', the fGCM level was 1.15±0.11 (Fig 2). Fig 3 represents median values and quartile ranges for each enrichment regime for both winter and summer seasons. Fig 4 represents median fGCM values for each individual leopard during summer and winter months across three categories, category 'A' (Fig 4A), category 'B' (Fig 4B), and category 'C' (Fig 4C).

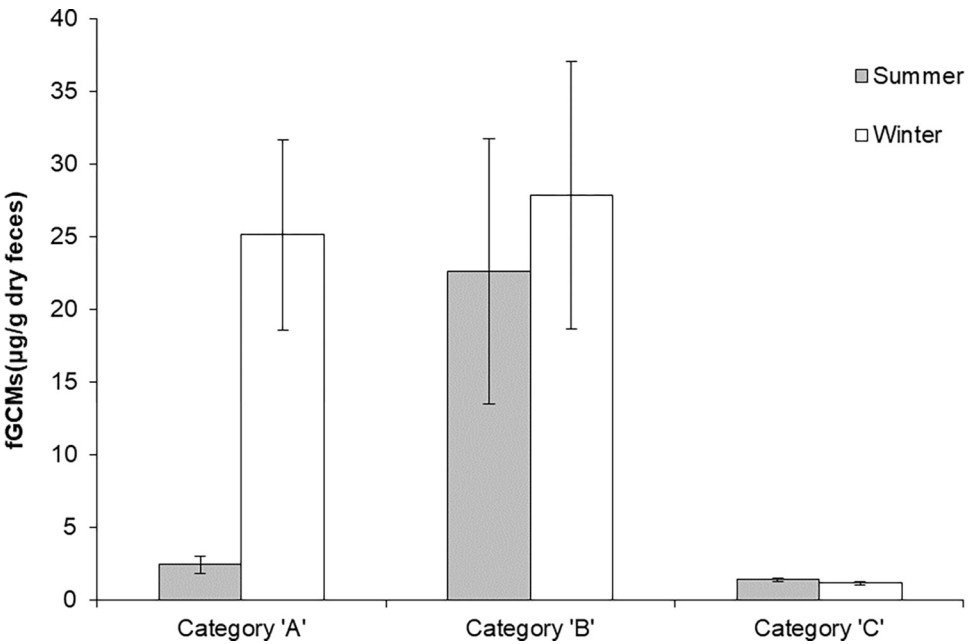

**Fig 2. fGCM levels (mean plot) of captive Indian leopards maintained under three different enrichment categories.** fGCM levels (Mean±SEM) in captive Indian leopards maintained under three different enrichment categories, 'A (active and passive enrichment)', 'B (active enrichment type I)' and 'C (active enrichment type II)', during summer and winter seasons.

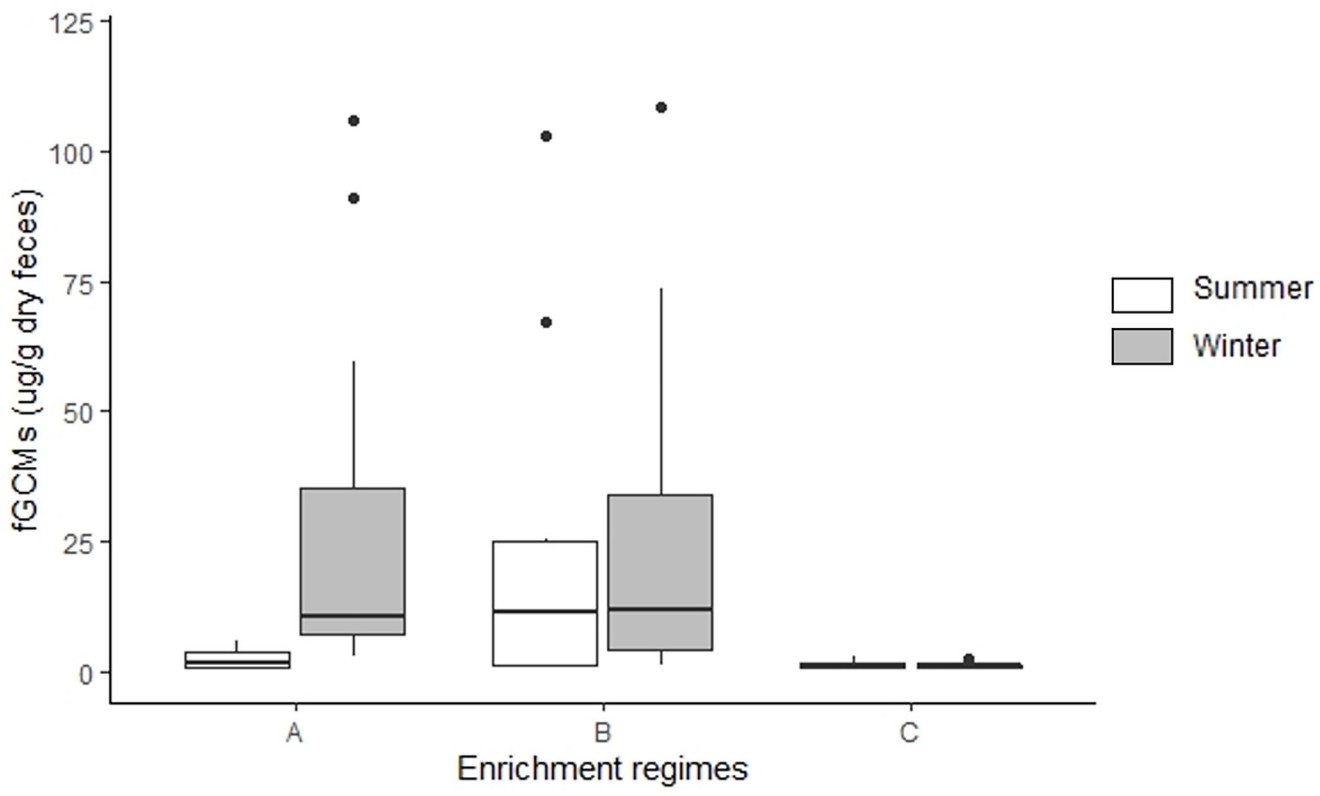

**Fig 3. fGCM levels (median plot) of captive Indian leopards maintained under three different enrichment categories.** fGCM levels (median levels with upper and lower quartile) in captive Indian leopards maintained under three different enrichment categories, 'A (active and passive enrichment)', 'B (active enrichment type I)' and 'C (active enrichment type II)' during summer and winter seasons.

### fGCM levels in captive leopards under different enrichment regimes during summer and winter seasons

A parent LME model was constructed using logfGCM~Enrichment*Season. Further comparison of LME models, with (logfGCM~Enrichment*Season) and without an interaction (logfGCM~Enrichment+Season) between enrichment and season, showed significant difference (P<0.001) (Table 3), and thus, interaction term was retained in the final LME model (Table 4).

Post-hoc Tukey's test across different enrichment categories showed significant differences in overall fGCM (pooled across seasons) concentrations between categories 'A' and 'C' (P<0.001) and between 'B' and 'C' (P<0.001), as well. No significant differences (P>0.05) were obtained between categories 'A' and 'B'. Further, there was a significant difference in overall fGCM levels (pooled across enrichment categories, 'A', 'B' and 'C') between summer and winter (P<0.01). Within group comparisons (grouped by enrichment category) of pairwise interactions showed significant differences in fGCM levels (Tukey post-hoc test, P<0.01) between summer and winter seasons for leopards in category 'A', and no significant differences (P>0.05) in fGCM levels between summer and winter seasons for both categories 'B' and 'C'. When grouped by season, Tukey's post-hoc test showed significant differences in fGCM levels between categories 'A' and 'B' (P<0.001), and categories 'B' and 'C' (P<0.001) during summer season, but no significant difference (P>0.05) was obtained between category 'A' and 'C' during the summer. In contrast, during winter season, significant differences

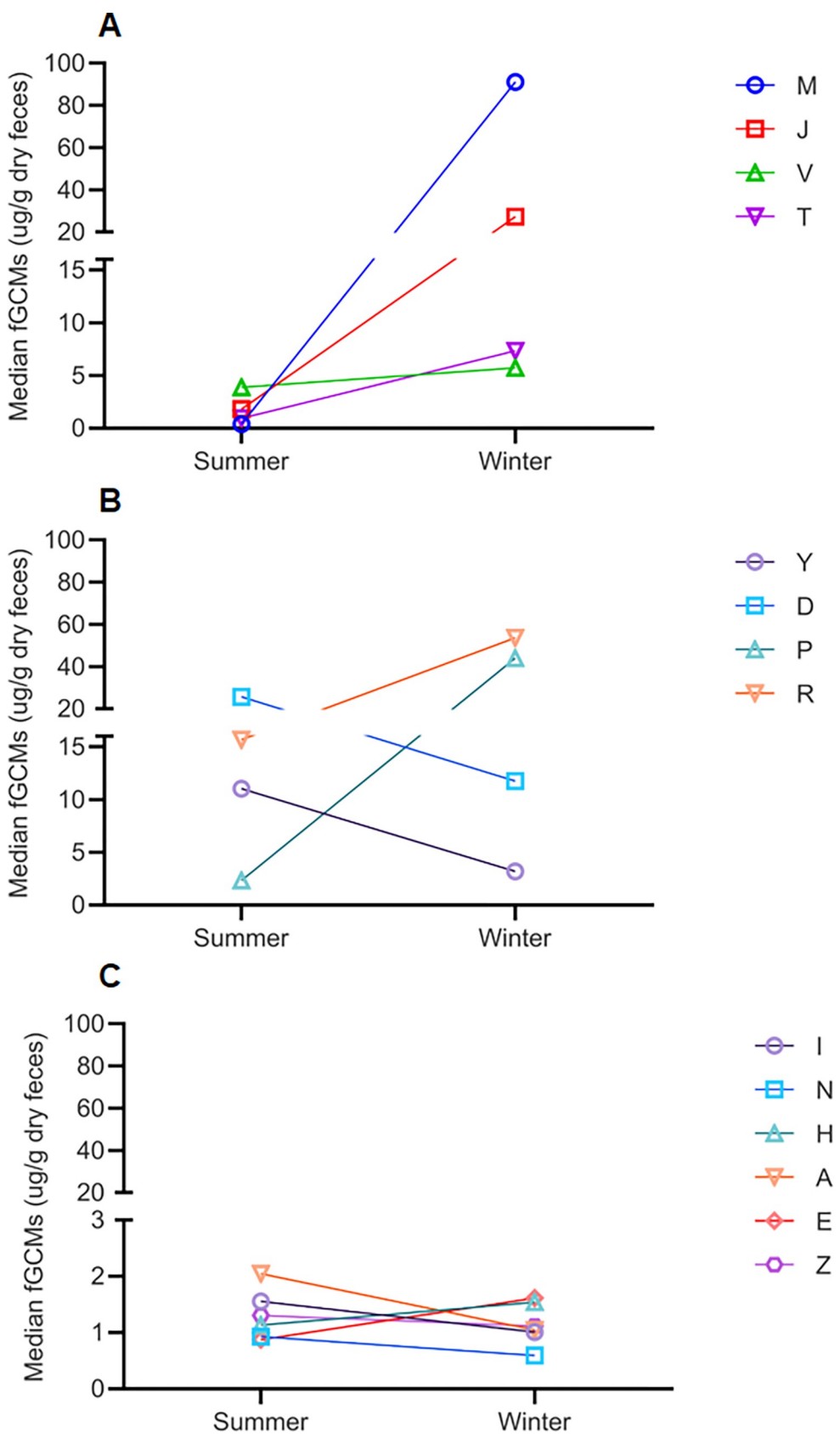

**Fig 4. fGCM levels (median plot) of individual captive Indian leopards maintained under three different enrichment categories.** fGCM levels (median values) in individual captive Indian leopards maintained under three different enrichment categories, A (active and passive enrichment)', 'B (active enrichment type I)' and 'C (active enrichment type II)' during summer and winter seasons. Panel A represents enrichment category 'A', panel B represents enrichment category 'B', and panel C represents enrichment category 'C'. A single color represents an individual indicated by an alphabet.

(Tukey post-hoc test, P<0.001) in fGCM levels existed between categories 'A' and 'C'. Significant difference (Tukey post-hoc test, P<0.001) was also obtained between 'B' and 'C' category-leopards during the winter season; however, there was no significant difference (Tukey post-hoc test, P>0.05) between 'A' and 'B' category-leopards for the winter season.

## fGCM levels of free-ranging leopards

A total of 12 samples was collected from three districts of Gujarat. Overall, fGCM concentrations of free-ranging leopards were 0.95±0.003 μg/g dry feces. The highest and the lowest fGCM values were 0.20 and 1.93, respectively.

## Parallelism curves

The slope of the regression lines for standard curve and the serially diluted samples (both captive and wild) (Fig 5) were not significantly different (P>0.05) and thus, the assay can be used to measure fGCM concentrations in both captive and wild Indian leopards.

## Discussion

Our study presents fGCM levels in Indian leopards housed in two Indian zoos and maintained under three different types of enrichment regimes, defined as active and passive enrichment category (category 'A'), active enrichment type-I (category 'B') and active enrichment type-II (category 'C'). Overall fGCM levels measured in our study were higher than those measured by Vaz *et al.* [26] in the scats of captive Indian leopards. However, Vaz *et al.* [26] utilized a specific corticosterone assay rather than a group-specific assay, which targets metabolites. A native hormone (glucocorticoid) is present only in trace amounts, if at all, in the feces [34]. Thus, using a native hormone assay for non-invasive samples, for example, feces, urine, is not appropriate and may lead to questionable results. Vaz *et al.* [26] measured corticosterone whereas cortisol is the predominant glucocorticoid released in mammals [35, 40, 41]. This explains the higher fGCM levels in our study when compared to Vaz *et al.* [26], and further, the assay used in their study [26] was not validated for leopards.

In our study, changes in adrenocortical activity in captive Indian leopards were detected by measuring a group of cortisol metabolites (with a 5α- 3β, 11β-diol structure), and the same assay has been validated and used for African leopards [13]. Thus, we directly compared fGCM levels in captive individuals between the African (*pardus* subspecies) and the Indian subspecies (*fusca* subspecies), and found the mean fGCM value to be 36 times higher in Indian

**Table 3. Comparison of linear mixed effects model with and without an interaction term.**

|  | df | AIC | BIC | logLik | Test | L.Ratio | p-value |
|---|---|---|---|---|---|---|---|
| Model1[a] | 8 | 341.07 | 363.30 | -162.5358 |  |  |  |
| Model2[b] | 6 | 366.79 | 383.46 | -177.3966 | 1 vs 2 | 29.72157 | <0.0001 |

[a]Model1:lme(logfGCM~Enrichment*Season,random = ~1|Ind.ID)

[b]Model2: lme(logfGCM~Enrichment+Season,random = ~1|Ind.ID)

**Table 4. Results from linear mixed effects model showing effects of enrichment categories, seasons and their interaction on fGCM levels in captive leopards.**

| Linear mixed-effects model fit by maximum likelihood | | | | | |
|---|---|---|---|---|---|
| **Random effects:** | | | | | |
| **Formula: ~1 | Ind.ID** | | | | | |
| **(Intercept) Residual** | | | | | |
| **StdDev: 0.1876867 0.3979013** | | | | | |
| **Fixed effects: logfGCM ~ Enrichment*Season** | | | | | |
| | **Value** | **Std.Error** | **F** | **t-value** | **p-value** |
| (Intercept) | 0.2024 | 0.1339 | 00 | 1.5112 | 0.0149 |
| ERB[a] | 0.7000 | 0.1893 | 00 | 3.6970 | 0.0068 |
| ERC[b] | -0.1561 | 0.1590 | 00 | -0.9814 | 0.4322 |
| SEASONWN[c] | 0.9611 | 0.1511 | 00 | 6.3590 | <0.0001 |
| ERB:SEASONWN | -0.7404 | 0.2272 | 00 | 3.2584 | 0.0013 |
| ERC:SEASONWN | -1.0395 | 0.1846 | 00 | -5.6295 | <0.0001 |

[a]ERB = Enrichment category 'B'

[b]ERC = Enrichment category 'C'

[c]SEASONWN = Winter season

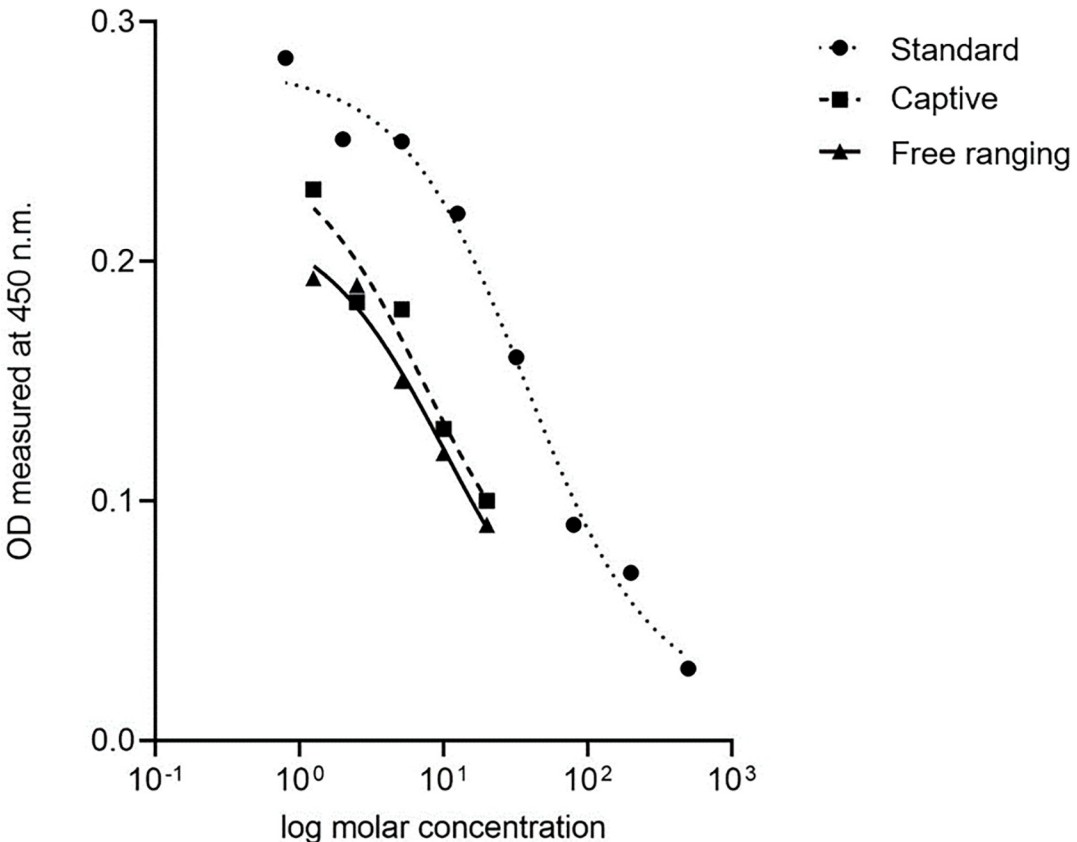

**Fig 5. Parallelism plots of fecal extracts with standard curve.** Parallelism plot showing serial dilution curves of extracted samples collected from free ranging and captive leopards, and the standard curve.

leopards than their African counterparts [13]. Such a huge difference in fGCM values between the Indian and the African subspecies can be due to multiple factors, ranging from differences in local environment to physiological differences at a subspecies level. For example, a study on white crowned sparrows (*Zonotrichia leucophrys pugetnesis*, *Zonotrichia leucophrys gambelli and Zonotrichia leucophrys nuttalli*) demonstrated that individuals breeding at higher altitude had increased stress hormone levels when compared to the subspecies, which bred at lower altitude [42]. In contrast, Gonzalez *et al.* [43] demonstrated subspecies-level differences in plasma cortisol concentrations for squirrel monkeys (*Saimiri sciureus sciureus* and *Saimiri sciureus albigena*) that were reared under identical laboratory conditions. In contrast, significant differences in fGCM levels were not observed between free-ranging Indian and African leopard subspecies. The fGCM values of free-ranging Indian (the current study) and African leopard subspecies [13] are quite overlapping. However, the sample size for free-ranging Indian leopards was significantly lower in our study. The scat samples collected from free-ranging leopards were for the purpose of validating the fGCM assay through generating parallelism curves, and any further insights on the fGCM patterns of free-ranging leopards will warrant future investigations.

Interestingly, there were no significant differences in fGCM levels between captive male and female leopards for both the African [13] and the Indian subspecies (the current study) (S1 Table). Sex-specific differences in fGCM levels were not so evident in other carnivores [44, 45], as well, except during the gestation phase. Jepsen *et al.* [45] showed that pregnant female tigers had significantly higher fGCM levels than males and non-pregnant females. The effect of sex on fGCM levels can be further influenced by several physical and social factors, for example, social rank, age, life history, and reproductive status [46]. In our study, the sampled leopards were all adults with no breeding history and were held in social isolation (one leopard per cage). Such lack of differences in physical and social make-up may possibly explain the similar fGCM profiles for both sexes in our study population.

The results from our study indicate a statistically significant effect of enrichment on the fGCM values of captive leopards. The leopards in active enrichment type I (category 'B') had the highest mean fGCM values (for both summer and winter seasons, and overall mean including both the seasons, as well) when compared to the other two categories. In contrast, leopards in active enrichment type II (category 'C') had the lowest mean (for both summer and winter seasons, and overall mean including both the seasons, as well) value of fGCM. Active enrichment type-I and -II (categories 'B' and 'C') were similar in quality having only elements of active enrichment (earthen floor, raised platforms). The only difference was that active enrichment type I (category 'B') had the smallest cage size and was provided with air coolers during summer. In contrast, active enrichment type II (category 'C') had medium size cages (48.94% larger than 'B') with no air coolers during summers, and the category having both active and passive enrichment (category 'A') had the largest cage size (78.07% larger than 'B'). Thus, the overall high fGCM levels of leopards in active enrichment type I (category 'B'), when compared to the other two conditions, indicated that cage size could be an important enrichment element for the given population. Similar results have been documented in other species, as well, where one of the elements had a stronger physiological effect than the rest within a given enrichment regime. Lapinski *et al.* [47] showed that cage size had no effect on salivary cortisol levels in foxes (*Vulpes vulpes*). However, gnawing sticks, another type of active enrichment, lowered the hormone levels of captive foxes, from 4.65±0.98 to 3.70±1.01 ng/ml. In this study, we did not measure other variables, for example, keeper's attitude towards the leopards, occurrence of diseases in captivity and the frequency of visitors, across different enrichment categories, which could possibly impact the fGCM levels and thus, needs further systematic analyses.

Studies have documented that the physiological effects of enrichment elements is quite diverse and usually varies from species to species, and even across different individuals of the same species. For example, a study on Asian elephants [48] showed differences in fGCM levels across individuals, though all were provided with similar kinds of structural enrichment. This pattern is also evident in our study, where conditions with both active and passive enrichment (category 'A) and active enrichment type I (category 'B') had large individual variations in fGCM levels. This was more evident in active enrichment type I (category 'B'), where individual variations were more prominent irrespective of the differences across seasons. Interestingly, active enrichment type I (category 'B') also had the highest mean fGCM values compared to the other two categories. Several studies have documented that individual variations in stress hormone levels are more prevalent when hormone levels increase under stressful conditions [49]. For example, a study in barn swallows [50] showed that with rise in predation pressure, GC levels increased, and this was also accompanied with a large degree of individual variations. Thus, to understand the physiological wellbeing of a target species, systematic monitoring at an individual level needs to be conducted.

Our study is the first of its kind to include passive enrichment elements in understanding the wellbeing of the captive leopards. Category 'A' was the only regime where passive enrichment elements, for example, natural sounds, sound proof glass to filter visitors' noise and artificially maintained lower ambient temperature, were provided along with active enrichment components. However, there were no significant differences in fGCM values between categories 'A' and 'C' during the summer, where 'C' is the regime with active enrichment elements only. Thus, at a physiological level, the combination of active and passive elements in category 'A' had a similar effect on fGCM levels when compared to active enrichment regime in category 'C'. To the best of our knowledge, no study has measured fGCM levels in leopards housed under conditions having both active and passive enrichment elements, and further, has compared the levels with conditions having active enrichment elements only.

We did not find any significant effect of seasons on fGCM levels, except for the leopards in conditions with both active and passive enrichment elements (category 'A'), where fGCM levels were higher during winter. However, we speculate that the higher fGCM levels in winter was not a 'season effect', but could be attributed to the presence of two old (>19 years) individuals in category 'A'. Both individuals showed high values during winter when compared to summer, and eventually died after a span of 8–10 months. Several studies have found that old age is associated with relatively higher fGCM levels when compared to the younger age classes [51–53]. Studies [51–53] have demonstrated that association between fGCM levels and age of the individual animal is very context-specific, and depends on several environmental and biological factors [51–53]. Thus, in our study, we speculate that the high fGCM levels in the two >19 years old individuals, when compared to other individuals under the same conditions with both active and passive enrichment elements (category 'A'), could be due to their older age, However, this is only a preliminary finding, which needs to be corroborated with long-term, longitudinal fGCM profiles of captive leopards across different life-history stages.

In conclusion, our study showed that enrichment, in particular size of the cage, influenced fGCM levels of captive Indian leopards within our sampling population. However, our study also revealed that physiological responses were quite diverse, showing huge variations in fGCM levels across individuals, and even for the same individual between summer and winter seasons. To accurately monitor physiological responses, zoo management programmes should focus on collecting several samples from multiple individuals, including equal numbers of males and females. Further, various covariates, including both individual- and environment-specific factors, should be included in the analyses to understand physiological wellbeing of the target species within a captive environment. Individual factors may include, sex ratios, age,

life-history stages, duration spent under captive conditions, and social rank, health and disease status of the animal. Environment-specific factors may include types of enrichment element, diet regime, presence of conspecific individuals, seasons, visitors' frequency, and keeper's attitude. Our study applied the same fGCM assay method for Indian leopards that has previously been validated for the African ones. Utilizing the same assay for different sub-species in geographically different regions allows for direct comparisons of endocrine profiles. In terms of management practices, the study provides a validated fGCM assay to monitor the wellbeing of Indian leopards, both under captive and free ranging conditions, and underlines the overall importance of cage size as an enrichment element for the sampled leopard population. The findings will contribute towards management and conservation of the Indian leopards.

## Supporting information

**S1 Table. Comparative overview of fecal glucocorticoid metabolite (fGCM) levels (Mean ±SEM), and value range; μg/g dry feces in Indian leopards from our data and in Indian and African leopards from published data.**
(DOCX)

## Acknowledgments

The authors would like to thank the zoo directors Shri. RK Sahu and Dr. Pratyush Patankar for giving the permission to collect scat samples from Kankaria and Baroda zoos, respectively. The authors also thank Shri. Shyamal Tikadar (Principal Chief Conservator of Forests, Forest Department, Gujarat, India) for providing permit to collect scats in protected areas, and Krunal Trivedi, Surat Nature Club for cooperation and support during sample collection from wild leopards. NP and CD acknowledges the Integrated master's programme of Ahmedabad University for providing an opportunity to conduct the research as a part of their master's dissertation project.

## Author Contributions

**Conceptualization:** Chena Desai, Ratna Ghosal.

**Formal analysis:** Ratna Ghosal.

**Methodology:** Nirali Panchal, Chena Desai, Ratna Ghosal.

**Project administration:** Nirali Panchal, Chena Desai, Ratna Ghosal.

**Visualization:** Ratna Ghosal.

**Writing – original draft:** Nirali Panchal, Chena Desai, Ratna Ghosal.

**Writing – review & editing:** Nirali Panchal, Chena Desai, Ratna Ghosal.

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
