## [Decision Letter · Decision Letter 0]

8 Feb 2022

PONE-D-21-38460Effects of active and passive enrichment regimes on fecal glucocorticoid metabolite levels in captive Indian leopards (Panthera pardus fusca)PLOS ONE

Dear Dr. Ghosal,

Thank you for submitting your manuscript to PLOS ONE. After careful consideration, we feel that it has merit but does not fully meet PLOS ONE’s publication criteria as it currently stands. Therefore, we invite you to submit a revised version of the manuscript that addresses the points raised during the review process.

As suggested three revewers and my own reading the manuscript is required a subtantial revision before it can be considered publication.

We look forward to receiving your revised manuscript.

Kind regards,

Govindhaswamy Umapathy, PhD

Academic Editor

PLOS ONE

Journal Requirements:

“The project was supported through Ahmedabad University’s start-up grant awarded to RG. NP”

“RG received startup grant from Ahmedabad University, Gujarat. The funders had no role in study design, data collection and analysis, decision to publish, or preparation of the manuscript.”

7. We note that Figure 1 in your submission contain map images which may be copyrighted. All PLOS content is published under the Creative Commons Attribution License (CC BY 4.0), which means that the manuscript, images, and Supporting Information files will be freely available online, and any third party is permitted to access, download, copy, distribute, and use these materials in any way, even commercially, with proper attribution. For these reasons, we cannot publish previously copyrighted maps or satellite images created using proprietary data, such as Google software (Google Maps, Street View, and Earth). For more information, see our copyright guidelines: http://journals.plos.org/plosone/s/licenses-and-copyright.

Reviewers' comments:

Reviewer's Responses to Questions

**Comments to the Author**

1. Is the manuscript technically sound, and do the data support the conclusions?

Reviewer #1: Yes

Reviewer #2: Partly

Reviewer #3: Yes

2. Has the statistical analysis been performed appropriately and rigorously? 

Reviewer #1: Yes

Reviewer #2: Yes

Reviewer #3: Yes

3. Have the authors made all data underlying the findings in their manuscript fully available?

Reviewer #1: Yes

Reviewer #2: No

Reviewer #3: Yes

4. Is the manuscript presented in an intelligible fashion and written in standard English?

Reviewer #1: No

Reviewer #2: Yes

Reviewer #3: Yes

5. Review Comments to the Author

Reviewer #1: The authors have provided a robust study that provides valuable information related to the housing and welfare of a cryptic and solitary carnivore in captivity. The manuscript meets the requirements for acceptance however I would encourage the authors to obtain assistance from a native English speaker to assist with grammar and sentence structure. Although the information is important it could be more clearly communicated. I therefore recommend acceptance of the manuscript for publication after major revision

Abstract

Line 12 – improves the

Line 13 – first mention of species in the Abstract should have (scientific name)

Line 14 – clarify interacting and non-interacting – are zoo keepers interacting with animals or are animals interacting or not interacting with certain aspects of their environment/enrichment mechanisms – perhaps use the term active and passive

Lines 16-20 – categories should be more specific – the difference between large, medium and small cage size provides no distinction in cage size for the reader

Line 28-29 – The authors need to clarify what policies this information will inform – do you refer to the management of captive, free-ranging leopards or both??

Introduction

I found the introduction difficult to get through, its also very long – with a bit of restructuring, the authors could build a clear and solid foundation that introduces the importance of their work. I would suggest restructuring the introduction and possibly involving an English speaker to assist with sentence structure, grammar and word order.

First introduce the importance of zoos for maintaining genetic integrity of populations and behavioural studies - Focus on India and your species of interest – move lines 118-139 up to the beginning of the intro

Then deal with the difficulties related to the management of zoo animals of different species – which can result in stereotypical behaviour and other negative effects - Then discuss stereotypical behaviours and results of confinement – introduce the concept of enrichment through different mechanisms – food, size of enclosure and define and reference active (Line 58-59) or passive (Line 67-69) enrichments.

Introduce the concept of stress and go into some detail related to stress physiology, which can reduce physiological health, cause reproductive issues or welfare issues in captive settings

Then highlight why it is important to be able to monitor stressors in the most effective, non-invasive and standardized way in this species to ensure optimal health of captive individuals and point out the focus of your study and how it bridges the gaps in our knowledge related to your specific species and their management in zoos in terms of monitoring stress

Line 50 – topmost is not a word

Line 57 – the authors state that studies have shown – but don’t reference these studies also no references are given for definitions of active and passive enrichment within the study setting

Line 60 – I do not understand the term “a study in literature” – are the authors referring to a literature review or are they referring to the fact that they did a literature search

Line 61 refers to ‘proper housing’ - for readers outside of a captive zoo setting, this provides no useful information – be specific in your description of what constitutes proper housing

Line 62 refers to studies conducted on various species but no references have been given for these studies

Line 62 and 75 no scientific names of species mentioned are included in the introduction

Line 63-67 is repetitive – “Bears had higher  locomotor and exploratory behaviours when food was provided in different ways, such as in a log filled with honey and by hiding food throughout the exhibit. Multiple ways of food  presentation helped bears to reduce stereotypic pacing from 125min/day to 20min/day, and  increased the rate of their exploratory behaviours [6].”

Reword to When food presentation was varied, such as in a log filled with honey and by hiding food throughout the exhibit, bears reduced stereotypic pacing from 125min/day to 20min/day, and increased the rate of their exploratory behaviours [6].

Line 81 “Long-term physiological stress is highly detrimental to zoo animals” it is also detrimental to free-ranging animals – I would focus the attention to zoo animals but not restrict your comments to captive individuals only. You also have no references for this and below statements related to the negative physiological effects of stress

Line 84 – definition of stress should be referenced

Distinctions between the stressor and the stress response should be more clearly defined that include physiological and psychological aspects

Line 85 – “A series of physiological events take place to restore this disruption is defined as the stress response” This does not make sense, the authors are saying that disruption (i.e. stress) is restored by physiological events – I think they mean that homeostasis is restored through the stress response –

Line 86-91 – this could be more clearly worded to describe the differences in extrinsic and intrinsic stressors. Utilize terminology suitable for animals – aggression not anger or refer to in humans or in captive animals for these stressors

Lines 91-95 – “Under stressful conditions, stress hormones are released as a biological response   helping to cope with “fight or flee” situations. Most kinds of stressors trigger the release of   glucocorticoid (GC) hormones [10], and GCs are known to increase blood glucose levels,  suppress immune system to help maintain homeostasis, and support metabolism of fats,   proteins and carbohydrates [11,12] to fight against the stressful situations.  ” I do not think this is an adequate description of the stress response – the authors do not clearly describe the physiological processes of the stress response, where GC come from, why they can provide a measure of stress or how they can be measured or that faeces is used in this study because it meets all of the criteria for non-invasive assessment.

Materials and Methods

This section is quite complicated to read and should be revised to more clearly outline the differences between the sites as well as the categories and number of animals sampled at each site.

I would suggest splitting the section into two section that deal with Captive and Free ranging or into three sections that deal with Captive and Free-ranging and then Sample collection – Either provide a summary of each site in words then add info (number of categories, number of animals, number of samples, etc.) in a table or provide details in text – there is no need to repeat information or leave information out.

If you provide details in the text, make it easier for the reader to go back and check number of categories at a site, or number of samples from an animal

I would start the captive section by describing the overall environmental conditions of the area (as zoos are quite close together) this will help to put any environmental differences between free-ranging and captive animals in context for the reader. Give the same information for each facility – currently you provide size of Kankaria zoo but not number of animals housed but for Sayajibaug zoo you give number of animals and species – Be consistent and be specific.

Then you provide information on each site specifically – I would keep things simple here

e.g. Leopards housed in both facilities are adults. In Kankaria/Kamala Nehru zoo, Ahmedabad, Gujarat indoor and outdoor housing is provided while at the Sayajibaug zoo only outdoor housing is offered. Provide further details in a comparative way regarding cage size, enrichments etc.

Then you describe sample collection methods within each setting – Currently (line 196) you provide total number of samples but do not state from which facilities these samples came from. I would revise Supplementary Table 1 and include it in your methods – or include this information in the text to make it easier for the reader to follow.

The authors state that only fresh faecal sample collection from animals in the wild occurred and that determination of freshness was based on moisture content, smell and state of decomposition. If you do not observe the animal defecating, how did you determine state of decomposition? The moisture content, smell and state of decomposition are all determined by the last meal – if the meal consisted of an old carcass, sinew and bone – moisture content, smell and visual content of the faeces will differ markedly to faeces resulting from a blood rich or flesh meal. The authors should consider rewording this to show that samples were considered fresh if “list your criteria” i.e. warm to the touch, fresh tracks visible, <12 hours etc.

I do not see any mention of why you use wet weight for some samples and dry weight for others – perhaps you can provide some info on this in your methods??

Statistical analysis

You do not include seasonal aspects as one of your stated aims of the project in your introduction or in your data collection sections but you mention it in the statistical analysis and results section – I would suggest making a clear link between possible influences of season on animal welfare in captive individuals and stress levels if this is something you are going to address in your results

You could also add a single sentence in your statistical analysis section indicating when fGCM are reported as micrograms /gram dry weight (DW) or wet weight (WW) instead of repeating this in your reporting of results for each variable.

Results

fGCM profile of captive leopards

If you add a sentence to your statistical analysis for reporting WW and DW – you do not need to state it after reporting each concentration

Your sample size is quite small, how do you account for this limitation in your statistical approach?

Discussion

Throughout the discussion you refer back to the different categories, it may be easier for the reader to follow if you outline the criteria from that category and then add the category in brackets after i.e. active and passive enrichment (Category A)

Lines 382 – 392 – the statement that sex-specific differences in cortisol are not prevalent in other carnivores is a bit misleading as you then state in the next sentence that reproductive state in females influences cortisol concentrations in tigers – the same was observed in the study on African leopards – I suggest rephrasing this to more clearly highlight higher cortisol levels in reproductive females

Line 387 – The effect of sex type is not correct – change to the effect of sex/gender

Line 395 – what do the authors mean by season-wise are you referring to per season?

Be careful of using the phrase “in the current study (Lines 351, 360, 374, 384, 389, 394, 411, 429, 440, 453, 463, 468, 470) and or /our study (349, 357, 378, 393, 419, 449, 455” too often – use only when comparing directly to another study

You sometimes refer to stress hormone metabolites (lines 397, 405) and then other times to fGCM concentrations/values (lines 395, 403) – remember your work might be of use to people outside the field of endocrinology so keeping your terminology consistent will help readers follow your story better

The use of “high” does not convey any quantitative information nor does it contextualise the level – please go through the discussion and reword to be more specific where applicable

Line 403 – you state high fGCM values of leopards in category B – high in comparison to what? Other categories, free ranging animals - you need to be specific

Line 420 high variation – in comparison to what?

Line 422 high levels in comparison to what?

Line 425

Line 426

Line 447

Line 450

Line 415 “Studies have documented that physiological effect “ should read Studies have documented that the physiological effect or physiological effects

Line 431 – should read “sound proof glass” not glasses and sentence should be in active voice “

i.e. Category ‘A’ was the only regime where passive-enrichment elements, for example, natural sounds, sound proof glass to filter visitor noise ….“

Lines 457 – 463 very long sentence, try breaking up into two

Reviewer #2: The paper submitted by Panchal et al to PLoS ONE reports fecal glucocorticoid metabolite (fGCM) levels in an exploratory study in captive Indian leopards. Animals were kept under three different housing conditions (categorized A-C), providing different types and degrees of enrichments. The authors utilized the same enzyme immunoassay, which has been previously fully validated for African leopards. Overall fGCM levels were much higher in captive animals compared to wild ones, from which a few samples were also included for comparison. Huge individual variations in fGCM levels in captive animals were found, but also differences between seasons, whereas sex did not affect levels. In addition, cage size influenced levels significantly.

Given the setting of the study (for obvious reasons no experiments could be performed), I think it is not possible to speak about “effects” of enrichment. In addition, in categories “A” and “B” only one sex was present, thus the influence of sex and enrichment in those two settings cannot be disentangled. I found the categories, “active” and “passive” enrichment interesting. However, I wonder, whether cage size is really an active enrichment (e.g. line 404)?

Below please find my specific comments. I thought it is easier for all involved to make my suggestions related to style or wording directly into the word file (track mode for changes).

Further detailed comments (ordered by appearance in the ms):

Line 1/2: I suggest rewording: “Fecal glucocorticoid metabolites in captive Indian leopards (Panthera pardus fusca) housed under different enrichment regimes”

Line 14: A (zoo) animal is not a “system”

Line 21 (and elsewhere): You did not “standardize” a non-invasive method. There is no standard available, all immunoassays give relative concentrations when applied to fecal samples (because there is a mixture of different metabolites present there). However, you need to validate such methods (biochemically, but more important physiologically/biologically).

Lines 63-67: I doubt that those details are necessary to describe here. Please delete.

Line 92: The classical “fight or flight” reaction is linked to the sympathetic nervous systems, and includes the release of catecholamines.

Line 103: I suggest citing a review here, at least instead of the primate paper (or why is this cited, and not other ones?)

Lines 127/128: Add the “metabolites”. Even when the original paper has it wrong, there are no glucocorticoids in the feces, only their metabolites.

Table 1: I think it would be better to move it to the supplement. Besides, the column “Breeding history does not give any additional information and should be deleted (also the “Remarks”). “Gender” should be replaced by “Sex” – it is about the biological term here.

Table 2: You give the cage side in m3, but if no details about “length*breadth*height” are given, the latter can be deleted.

Lines 204 and 218: You state that four districts were chosen, but then samples were only collected from three?

Line 215: What accounts for 7.4%, unclear to me?

Lines 237-252: This section needs drastic revision. I have made a suggestion directly in the docx-file. You need to cite the original paper, where the EIA was first described (Touma et al., 2003). 5α-pregnane-3β,11β,21-triol-20-one is the standard of this EIA. However, as this is a corticosterone metabolite, but cortisol is the dominant GC in carnivores, it is most likely not present in the feces. Still, because the EIA picks up metabolites with a 5α-3β,11β-diol structure, which can also be derived from cortisol (respective cross-reactions have only recently been described in Santamaria et al., 2021), this EIA has been proven suited in African leopards [9].

Line 259: Well, category and sex were not independent, because category A had only females and “B” only males.

Results: Overall, I think it is more adequate to present levels as median with range (min-max), and show them in respective figure 2 as boxplot graphs. The variation was so huge (sometimes the SE is greater than the mean) and thus data not normally distributed, or?

Line 287: mean±SE?

Line 287 onwards: “μg/g dry wt of feces” – I think you can state somewhere that fGCM levels are expressed as μg/g per dry weight of the feces and then reduce that to “μg/g feces” throughout.

Line 296 – Figure 2: As the SE is huge (sometimes more than the mean), I think it’s better to give boxplot graphs of the values than presenting mean±SE.

Line 300: For reason outlined above (“effects”), I suggest to delete this heading (it should not have the same level as the one above (line 286)

Table 4: a p value of 0.0000 is impossible, needs to be <0.0001.

Lines 344-347: Move this to the supplements. It is only a small technical details and not of general interest (or importance).

Lines 349-359: Vaz et al (2017) utilized a corticosterone immunoassay. However, this does not mean that they measured corticosterone in the feces (see e.g. Palme, 2019 for a more detailed explanation). They did not characterize the immunoreactive compounds in the fecal samples of Indian leopards. Because glucocorticoids are heavily metabolized prior to excretion and also (as you correctly stated) cortisol and not corticosterone is the predominant GC, it is very likely that their assay picks up (cortisol) metabolites. As cross-reactivity with those metabolites is expected to be low, this perfectly explains the lower fGCM levels reported in [21]. I started to rewrite this paragraph, but I think more needs to be done here. In addition, the assay of Vaz et al. was not validated for leopards, which also warrants mentioning somewhere.

Line 355: I suggest to delete [32, 33], the first does not seem suited at all, and the second is included in the review [34].

Line 361: Actually, 5α-pregnane-3β,11β,21-triol-20-one is a corticosterone metabolite (the 17α-OH, the only difference between cortisol and corticosterone, is missing). However, because the immunogen for the antibody, and the label of this EIA have been coupled at position C20, the assay cross-reacts with both, cortisol and corticosterone metabolites. This was recently proven (see Santamaria et al., 2021).

Line 387: “very variable”: Are there studies available, which report sex differences in a carnivore species? Otherwise this sentence does not make sense.

Lines 423/424: This sentence sounds odd. Please reword.

Lines 438-439: I would be very cautiously in drawing conclusions here. You did not perform experiments, just compared different housing situations. To answer such questions I think it would be necessary to plan longitudinal experiments with the same animals and provide different enrichments.

Line 445: So the two old animals were no longer present in summer?

Lines 457-463: Attention! Suggesting to include a huge number of covariates will decrease statistical power, which will result in more negative findings (no effect found), when sample numbers are not increased at the same time. Thus, I think the first demand should be to increase the number of individuals studied and the number of samples collected!

There is one further study published, which investigated fGCM levels in a single captive Afghan and black leopard each, before, during, and after the period of exhibit construction (Chosy et al., 2014). The authors used a corticosterone and a cortisol EIA, respectively, and found higher levels during construction (up to ~1 µg/g feces). It may be worth including this paper in the discussion.

References: The references need careful revision (e.g. capitalized names in 5; first names spelled out in 41) – why are internet links (doi) given for some papers, but not others? Latin names should be in italics, etc…

Figure 1: The resolution of my copy was rather low (but I hope the original one is better), thus it was hard to get the details.

Figure 2: I suggest presenting the levels as boxplot graphs (because variation was so huge; and data not normally distributed, or?). Comparing levels here, with data given in the “Results” sections (e.g. line 291 for the winter in “A”: 25.11±29.39 µg/g) make me wonder, why they do not match (the error bar should exceed the box)? Y-axes legend: “fGCMs (µg/g dry feces)”

Figure 3: It would be easier to orient oneself, when different colours and also open symbols are used. Also may be better to give median levels, instead of mean. The y-axes should then read: “median fGCMs (µg/g feces)”

Figure 4: As outlined above, I suggest moving it to the supplementary material. And it’s not the sample that is “wild” or “captive” �.

Supplemental Information (for review purpose only): I encourage the authors to make that information available in a supplement (especially the number of samples is interesting to know). Also, it may be worth noting (remarks) that Vaz et al did not validate their EIA for use in Indian leopards.

Above cited references:

Chosy, J., Wilson, M., Santymire, R. (2014): Behavioral and physiological responses in felids to exhibit construction. Zoo Biol. 33, 267-274. https://doi.org/10.1002/zoo.21142

Palme, R. (2019): Non-invasive measurement of glucocorticoids: advances and problems. Physiol. Behav. 199, 229-243. https://doi.org/10.1016/j.physbeh.2018.11.021

Santamaria, F., Barlow, CK., Schlagloth, R., Schittenhelm, RB., Palme, R., Henning, J. (2021): Identification of koala (Phascolarctos cinereus) faecal cortisol metabolites using liquid chromatography-mass spectrometry and enzyme immunoassays. Metabolites 11, 393. https://doi.org/10.3390/metabo11060393

Touma, C., Sachser, N., Möstl, E., Palme, R. (2003): Effect of sex and time of day on metabolism and excretion of corticosterone in urine and feces of mice. Gen. Comp. Endocrinol. 130, 267-278. https://doi.org/10.1016/s0016-6480(02)00620-2

Reviewer #3: The manuscript is scientifically well-conceived and experimental design strategy to examine the environmental enrichment measures was good. Measures of fGCM as a read out to assess the physiological well-being is interesting. The study outcomes provide improvising opportunities in the zoo management practices and findings are important in the context of strategic development of captive management of big cats, Indian leopards, with implications in their conservation.

6. PLOS authors have the option to publish the peer review history of their article (what does this mean?). If published, this will include your full peer review and any attached files.

Reviewer #1: No

Reviewer #2: No

Reviewer #3: No

---

## [Author Response · Author response to Decision Letter 0]

12 Apr 2022

Dear editor and the reviewers,

Thank you for considering our manuscript. We have addressed all the concerns raised by the editors and the reviewers in the revised version of our manuscript. Please refer to the response to reviewers document for further details.

Sincerely,

Ratna Ghosal

---

## [Decision Letter · Decision Letter 1]

2 Jun 2022

PONE-D-21-38460R1Measurement of fecal glucocorticoid metabolites levels in captive Indian leopards (Panthera pardus fusca) housed under different enrichment regimesPLOS ONE

Dear Dr. Ghosal,

Thank you for submitting your manuscript to PLOS ONE. After careful consideration, we feel that it has merit but does not fully meet PLOS ONE’s publication criteria as it currently stands. Therefore, we invite you to submit a revised version of the manuscript that addresses the points raised during the review process.

Need to revise English language throughout the manuscript as one of the reviewer's has raised a serious concern before it can be considered for publication. 

We look forward to receiving your revised manuscript.

Kind regards,

Govindhaswamy Umapathy, PhD

Academic Editor

PLOS ONE

Reviewers' comments:

Reviewer's Responses to Questions

**Comments to the Author**

1. If the authors have adequately addressed your comments raised in a previous round of review and you feel that this manuscript is now acceptable for publication, you may indicate that here to bypass the “Comments to the Author” section, enter your conflict of interest statement in the “Confidential to Editor” section, and submit your "Accept" recommendation.

Reviewer #1: (No Response)

Reviewer #2: (No Response)

2. Is the manuscript technically sound, and do the data support the conclusions?

Reviewer #1: Yes

Reviewer #2: Yes

3. Has the statistical analysis been performed appropriately and rigorously? 

Reviewer #1: Yes

Reviewer #2: Yes

4. Have the authors made all data underlying the findings in their manuscript fully available?

Reviewer #1: Yes

Reviewer #2: Yes

5. Is the manuscript presented in an intelligible fashion and written in standard English?

Reviewer #1: No

Reviewer #2: Yes

6. Review Comments to the Author

Reviewer #1: Dear Authors,

The revised manuscript addresses many of the comments and suggestions set out by the reviewers, however one of the fundamental issues you have failed to address is the standard of writing. Your findings have merit and would be of value to other conservation entities and practitioners for the care and welfare of captive leopards, however you have not devoted the same time and effort into accurately communicating your results in grammatically correct English. This oversight detracts substantially from your study. Even your title is grammatically incorrect and I therefore encourage you once again, as I did in my first review, to seek the assistance of a native English speaker or to make use of the services of a professional copyeditor.

Reviewer #2: Thanks to the authors for substantially modifying their manuscript. It is much improved now, and I’m happy with it (also their responses to my suggestions). There are only a few, marginal things left, which I kindly ask the authors to correct/modify (see below):

General: Although you describe that levels are expressed “μg/g feces” somewhere in the methods section, it is still necessary to have the dimension when you report values somewhere (for the ease of the reader, but also correctness). You may avoid repetition in a sentence where several concentrations are given, but would need it at least once.

Line 4: delete “levels” it’s obvious and included in “measurement” – otherwise it should read: “fecal glucocorticoid metabolite levels”.

Line 96: [11] – unsure, but I think that this article only marginally deals with sample materials. What about replacing it with Sheriff et al., 2011 – there pros and cons of the different sample matrices are discussed in detail.

Line 280: µg!

Line 319: <.0001 add the 0  <0.0001

Line 347: A total of 12 samples was collected.

Line 365: [39] Where do they mention this? I suggest citing [33] here instead.

Line 372: “were detected”

Line 428: effects

Line 472: I suggested rewording, but it’s still here: “demonstrate” is too strong, especially as you did not perform any experiments to prove this – your study is only observational.

Sheriff, MJ., Dantzer, B., Delehanty, B., Palme, R., Boonstra, R. (2011a): Measuring stress in wildlife: techniques for quantifying glucocorticoids. Oecologia 166, 869-887. https://doi.org/10.1007/s00442-011-1943-y

7. PLOS authors have the option to publish the peer review history of their article (what does this mean?). If published, this will include your full peer review and any attached files.

Reviewer #1: No

Reviewer #2: No

---

## [Author Response · Author response to Decision Letter 1]

28 Jun 2022

Uploaded a seperate "Response to reviewers' file

---

## [Decision Letter · Decision Letter 2]

16 Aug 2022

PONE-D-21-38460R2Fecal glucocorticoid metabolite levels in captive Indian leopards (Panthera pardus fusca) housed under three different enrichment regimesPLOS ONE

Dear Dr. Ghosal,

Thank you for submitting your manuscript to PLOS ONE. After careful consideration, we feel that it has merit but does not fully meet PLOS ONE’s publication criteria as it currently stands. Therefore, we invite you to submit a revised version of the manuscript that addresses the points raised during the review process.

As suggested by Reveiwer,2, authors can revise the manuscript, before it can be accepted. 

We look forward to receiving your revised manuscript.

Kind regards,

Govindhaswamy Umapathy, PhD

Academic Editor

PLOS ONE

Journal Requirements:

Reviewers' comments:

Reviewer's Responses to Questions

**Comments to the Author**

1. If the authors have adequately addressed your comments raised in a previous round of review and you feel that this manuscript is now acceptable for publication, you may indicate that here to bypass the “Comments to the Author” section, enter your conflict of interest statement in the “Confidential to Editor” section, and submit your "Accept" recommendation.

Reviewer #1: (No Response)

Reviewer #2: All comments have been addressed

2. Is the manuscript technically sound, and do the data support the conclusions?

Reviewer #1: Partly

Reviewer #2: Yes

3. Has the statistical analysis been performed appropriately and rigorously? 

Reviewer #1: Yes

Reviewer #2: Yes

4. Have the authors made all data underlying the findings in their manuscript fully available?

Reviewer #1: Yes

Reviewer #2: Yes

5. Is the manuscript presented in an intelligible fashion and written in standard English?

Reviewer #1: No

Reviewer #2: Yes

6. Review Comments to the Author

Reviewer #1: Although somewhat improved from previous versions, this version still contains grammatical errors and the use of language that is not appropriate for scientific writing. See some examples below. As mentioned in previous revisions of this MS, the information is worthy of publication and is relevant for captive population measurement, however, the information needs to be communicated more clearly.

Line 80: Stressful conditions may also impact animal “emotions”

Emotions is an anthropomorphic term – rather use behavior

Line 81 increased aggression or increased repulsion under such conditions – Can the authors clarify what they are trying to say with this sentence

If they are referring to animal response related to a stressor and they are referring to behaviour – the authors should use the correct terminology – increased aggression or show of submissive behavior.

Line 89: poor reproductive performances – change to performance

Lines 93-95: Thus, to improvise ex-situ conservation efforts, enriched habitats are provided to captive animals and their physiological response towards the enrichment can be assessed through measurements of stress hormones, mostly by monitoring levels of GCs

Can the authors clarify what they mean by “improvise ex-situ conservation efforts

My understanding of the research, is that the authors used faecal samples and fGCM measurements to quantify stress in relation to different passive and active enrichments but here the authors talk about measuring stress hormones, mostly by monitoring levels of GCs as if they are measuring the biologically active steroid hormone for glucocorticoid measurement in the blood – The authors should clarify this please

Reviewer #2: (No Response)

7. PLOS authors have the option to publish the peer review history of their article (what does this mean?). If published, this will include your full peer review and any attached files.

Reviewer #1: No

Reviewer #2: No

---

## [Author Response · Author response to Decision Letter 2]

20 Aug 2022

We have submitted a separate 'Response to Reviewers' document.

---

## [Editor Report · Decision Letter 3]

24 Aug 2022

Fecal glucocorticoid metabolite levels in captive Indian leopards (Panthera pardus fusca) housed under three different enrichment regimes

PONE-D-21-38460R3

Dear Dr. Ghosal,

We’re pleased to inform you that your manuscript has been judged scientifically suitable for publication and will be formally accepted for publication once it meets all outstanding technical requirements.

Kind regards,

Govindhaswamy Umapathy, PhD

Academic Editor

PLOS ONE
---

## [Editor Report · Acceptance letter]

30 Aug 2022

PONE-D-21-38460R3 

Fecal glucocorticoid metabolite levels in captive Indian leopards *(Panthera pardus fusca)* housed under three different enrichment regimes 

Dear Dr. Ghosal:

I'm pleased to inform you that your manuscript has been deemed suitable for publication in PLOS ONE. Congratulations! Your manuscript is now with our production department. 

Kind regards, 

on behalf of

Dr. Govindhaswamy Umapathy 

Academic Editor

PLOS ONE